# S-methyl Methanethiosulfonate: Promising Late Blight Inhibitor or Broad Range Toxin?

**DOI:** 10.3390/pathogens9060496

**Published:** 2020-06-22

**Authors:** Charlotte Joller, Mout De Vrieze, Aboubakr Moradi, Claudine Fournier, Delphine Chinchilla, Floriane L’Haridon, Sebastien Bruisson, Laure Weisskopf

**Affiliations:** 1Department of Biology, University of Fribourg, 1702 Fribourg, Switzerland; charlotte.joller@unifr.ch (C.J.); mout.devrieze@unifr.ch (M.D.V.); aboubakr.moradi@unifr.ch (A.M.); delphine.chinchilla@unifr.ch (D.C.); floriane.lharidon@unifr.ch (F.L.); sebastien.bruisson@unifr.ch (S.B.); 2Medical and Molecular Microbiology, University of Fribourg, 1702 Fribourg, Switzerland; claudine.fournier@unifr.ch

**Keywords:** bacterial volatiles, MMTS, *Phytophthora infestans*, biofungicide

## Abstract

(1) Background: S-methyl methanethiosulfonate (MMTS), a sulfur containing volatile organic compound produced by plants and bacterial species, has recently been described to be an efficient anti-oomycete agent with promising perspectives for the control of the devastating potato late blight disease caused by *Phytophthora infestans.* However, earlier work raised questions regarding the putative toxicity of this compound. To assess the suitability of MMTS for late blight control in the field, the present study thus aimed at evaluating the effect of MMTS on a wide range of non-target organisms in comparison to *P. infestans*. (2) Methods: To this end, we exposed *P. infestans*, as well as different pathogenic and non-pathogenic fungi, bacteria, the nematode *Caenorhabditis elegans* as well as the plant *Arabidopsis thaliana* to MMTS treatment and evaluated their response by means of in vitro assays. (3) Results: Our results showed that fungi (both mycelium and spores) tolerated MMTS better than the oomycete *P. infestans*, but that the compound nevertheless exhibited non-negligible toxic effects on bacteria, nematodes and plants. (4) Conclusions: We discuss the mode of action of MMTS and conclude that even though this compound might be too toxic for chemical application in the field, its strong anti-oomycete activity could still be exploited when naturally released at the site of infection by plant-associated microbes inoculated as biocontrol agents.

## 1. Introduction

Potato late blight caused by the oomycete *Phytophthora infestans* is one of the most devastating diseases threatening potato crops worldwide, leading to tremendous monetary losses each year [1]. Despite rigorous monitoring of the spread, implementation of strict spraying regimes, the development of resistant cultivars and new pesticides, containment of the disease remains a major challenge for modern agriculture. This is even more true for organic farming, where options are mainly limited to copper-based fungicides and cultivation of more resistant cultivars. However, so far resistance breeding efforts were unsuccessful in developing durable resistant varieties and copper accumulates in soil and water, creating a toxic environment for microorganisms, flora and fauna [2,3,4]. Alternatives are thus urgently needed. In this regard, the development of efficient bio-control agents has proven to be a promising approach. Indeed, replacing pesticides and copper with microorganisms able to antagonize pathogens and promote plant growth, coupled with the use of resistant cultivars, could render agriculture more sustainable whilst maintaining high yields [5]. One often recurring challenge of biological control-based strategies is the need for the introduced organism to successfully establish in high population densities. Exploring active compounds derived from bacterial control agents for pathogen containment could therefore offer an alternative to the introduction of the biocontrol agents themselves.

Many studies have analyzed bacterial and fungal strains for their capability to produce diffusible antimicrobial compounds, yet only more recently focus has also been devoted to the high diversity of volatile organic compounds (VOCs) produced by plant microbiomes. Meanwhile, many VOCs have been described to efficiently inhibit plant pathogens [6]. Interestingly, Hunziker and colleagues observed *P. infestans* to be particularly responsive to VOCs emitted by bacteria naturally associated with potato plants [7]. A later in vitro screen of such volatiles identified several sulfur containing VOCs as exceptionally efficient inhibitors of the oomycete [8]. Subsequently, special attention was devoted to the sulfur containing S-methyl methanethiosulfonate (MMTS). This compound was shown to not only efficiently inhibit different life stages of *P. infestans* in vitro but also late blight symptoms in planta with seemingly low phytotoxicity when applied concomitantly to the oomycete and to restrict the pathogen growth also when applied after *P. infestans* infection [8,9]. MMTS was thus put forward as an attractive candidate for late blight control in the field.

In older studies, some attention had already been given to MMTS as part of the putatively health-promoting metabolome of *Brassicaceae* and *Alliaceae* plant species [10,11]. Produced in varying amounts by *Brassicaceae* and *Alliaceae* upon wounding and being of antimicrobial character, MMTS was proposed to be a kind of phytoalexin emitted by wounded plants to counterattack microbes [11,12]. In addition to its antimicrobial action, MMTS was shown to have antimutagenic features in *Escherichia coli* [11] and *Drosophila melanogaster* [13]. Furthermore, the closely related S-methyl methanethiosulfinate inhibited chemically induced genotoxicity in mice [14]. MMTS has moreover been studied as a degradation product of the insecticide methomyl, with a particular focus on its potentially toxic effects, i.e., toxicity on microorganisms and genotoxicity. Antimicrobial activity was also attested in this context but no genotoxicity on *Saccharomyces cerevisiae*, *Nicotiana tabacum* or *Salmonella typhimurium* were detected [15]. Finally, in 1984 Hoppf and colleagues, interested in MMTS for medicinal application, observed major toxicity in animals and determined a lethal dose median (LD_50_) value of 9.11 for intraperitoneal injection of MMTS in mice [16]. This is the only currently available estimation of the LD_50_ on a mammal for MMTS and it suggests rather severe acute toxicity, even though volatile exposure to MMTS is likely to be of less severe consequence than intraperitoneal injection.

Considering on the one hand these earlier reports of effects ranging from health-promotion to toxicity, and on the other hand the remarkable potential of MMTS to inhibit late blight *in planta*, the present study aimed to analyze the effect of this sulfur volatile on a wide range of non-target organisms, in order to evaluate its suitability for late blight control in the field.

## 2. Results

### 2.1. Effect of MMTS on Fungi in Comparison to its Effect on P. infestans

The effect of MMTS on mycelial growth of different fungi was evaluated compared to *P. infestans*. Oomycete and fungi were incubated on their respective growth media containing MMTS at different concentrations. The fungal species tested were chosen to include Basidiomycetes (*Rhizoctonia solani*, *Trichoderma* species) and Ascomycetes (*Botrytis cinerea*) as well as plant pathogens (*R. solani*, *B. cinerea*) and bio-control agents (*Trichoderma harzianum*, *Trichoderma artroviride*).

After short time exposure to MMTS (7 days), a clear dose-dependent effect was observed on *P. infestans* mycelial growth (Figure 1a): a slight inhibition was already visible at the lowest concentration tested, from 10 µg/mL on, the oomycete did not grow at all. When grown over a longer time period (14 days), the previously visible growth inhibition pattern faded, as the oomycete managed to overcome initial inhibition at lower concentrations (Figure 1b). At 10 µg/mL and 33 µg/mL however, the oomycete remained entirely inhibited up to 14 days of growth period. An efficient dose median (ED_50_) value of 2.4 µg/mL was calculated for the inhibition of *P. infestans* mycelium by MMTS by fitting a log logistic model to the data.

Similarly to *P. infestans*, fungi showed reduced/delayed mycelial growth at 33 µg/mL MMTS after short time exposure (Figure 1a). However, this reduction was much less pronounced than for the oomycete. Thus at 10 µg/mL of MMTS, where total growth inhibition of *P. infestans* was observed, *B. cinerea*, *R. solani*, *T. harzianum* and *T. artroviride* were only partially inhibited. Furthermore, in contrast to *P. infestans*, when grown over a longer time period, the fungi overcame initial inhibition almost completely at all concentrations tested, although some (e.g., *R. solani*) took longer than others (e.g., *T. artroviride*) to do so at the highest concentration (Figure 1b). No ED_50_ values could therefore be calculated for fungal inhibition. In general, no striking differences were observed between fungal species. Whether basidiomycetes or ascomycetes, bio-control agent or pathogen—fungi were all affected comparably by MMTS, yet to a much lower extent than the oomycete *P. infestans* (Figure 1, Appendix A).

To test whether this difference in sensitivity might also be due to different growth media, the same experiment was repeated with *B. cinerea* and *T. harzianum* growing on V8—the growth medium used for the oomycete—instead of PDA. Even though the fungi exhibited different morphology on V8 medium, their response to MMTS did not change substantially (Appendix A), and they even tolerated MMTS treatment slightly better when grown on V8 rather than PDA (Appendix A).

To investigate whether the differential susceptibility of fungi and oomycete observed for mycelial growth also held true for other developmental stages—such as propagating structures—the inhibitory effect of MMTS on *B. cinerea* spores was compared to that of *P. infestans* sporangia and zoospores. This was done by co-inoculating the different spores with different concentrations of MMTS and by monitoring their germination and survival.

Of the three structures tested, *P. infestans* zoospores were the most susceptible ones to MMTS treatment: propidium iodide staining—revealing dead cells—showed a significant increase of dead cells compared to the control at 50 ng/mL (Figure 2c) and at high MMTS concentrations zoospores appeared wrinkly, indicating that they might have been lysed. Moreover, motility and germination were entirely lost at 50 ng/mL of MMTS (data not shown).

When comparing the sensitivity of *P. infestans* sporangia with that of *B. cinerea* spores, we observed that *P. infestans* sporangia were more susceptible to MMTS than *B. cinerea* spores: sporangia did not germinate at 5 µg/mL and propidium iodide staining indicated high mortality at that concentration (Figure 2a). It should be noted however that *B. cinerea* germination was delayed compared to the control at 5 µg/mL and in some cases germinating spores where observed to form clusters at that concentration (Figure 2f).

Compared to *P. infestans* zoospores, *P. infestans* sporangia and *B. cinerea* spores tolerated MMTS treatment better. Thus, sporangia germination was completely inhibited at 5 µg/mL and germination of *B. cinerea* spores at 50 µg/mL (data not shown)—concentrations 100 and 1000 times higher than that required for total inhibition of zoospore germination.

### 2.2. Effect of MMTS on Eukaryotic and Prokaryotic Unicellular Organisms

The results described above show that fungal species are affected by MMTS—even if to a lesser extent than *P. infestans*—in mycelial growth and spore germination. To investigate the antifungal activity of this compound further, the direct inhibitory effect of MMTS against unicellular fungi was investigated by exposing Bakers’ yeast liquid cultures to different concentration of MMTS. The growth of yeast was significantly delayed at 7.8 µg/mL MMTS concentration, and total carrying capacity k of the culture was considerably lower than for the control. This did however not reflect into a drastically lower growth rate constant r (Figure 3a, Appendix A). Still, analysis of the integrated area under the growth curve, calculated by fitting a log-logistic model to the data, showed that already at 3.9 µg/mL MMTS, the growth was significantly different from the control. At 15.6 µg/mL MMTS and beyond, growth was completely inhibited. An ED_50_ value of 9.08 µg/mL was obtained by fitting a dose response curve to the empiric area under the growth curve. 

In addition to yeast, the effect of MMTS on growth in liquid culture against bacterial species was evaluated. A comparison between the inhibitory effect exerted by MMTS and kanamycin on the plant pathogenic bacterium *D. solani* showed that MMTS interferes with the growth of this bacterium in a concentration range comparable to kanamycin (Figure 3b and Appendix A). A further screen for growth inhibition by MMTS was carried out on bacterial species of clinical relevance carrying different antibiotic resistance markers and including both Gram negative and Gram positive species. This revealed that all bacterial species—irrespective of antibiotic resistance or cell wall composition—were affected by MMTS at minimal inhibiting concentrations (MIC) ranging from 16–32 µg/mL or in a few cases 64 µg/mL (Appendix A).

### 2.3. Effect of MMTS on Nematodes

Since MMTS affected not only fungal growth and the growth of *P. infestans*, but also bacterial growth—thus exhibiting a rather broad range of action—its effect on the development of the soil nematode *C. elegans* was examined. To that end, *C. elegans* L1 stage larvae where exposed to different concentrations of MMTS in liquid buffer containing *E. coli* OP50 as a food source and—as read-out of MMTS effect—the number of emerging L4 larvae after 48 h were counted. The number of surviving worms was dramatically low at 10 µg/mL MMTS and no alive animals were observed at ≥100 µg/mL (Figure 4). Functional analysis revealed that at concentration of 10 µg/mL the larvae were arrested in stages 1 and 2 and were not able to develop in further steps. At 1 µg/mL however only a slight decrease in survivors and normal development of worms was observed. An LD_50_-value of 2 µg/mL was estimated by fitting a log-logistic model to the data.

### 2.4. Effect of MMTS on Plants

Finally, in-vitro *A. thaliana* seedlings were exposed to MMTS in synthetic growth medium to determine in what concentration range the compound induces symptoms in plants. Interestingly, *A. thaliana* seedlings showed a similar pattern in their response to MMTS as had previously been observed for fungal mycelial growth. Thus, after short growth period on 10 µg/mL and beyond of MMTS-containing growth medium, seedlings were halted in growth—measured by root length—compared to the control. This initial inhibition however was entirely overcome after longer growth period at 10 µg/mL and partially also at 33 µg/mL MMTS (Figure 5). Even though inhibition symptoms—including stunted growth and chlorosis—remained at 33 µg/mL, seedlings started to make secondary roots and new leaves, suggesting that toxic effects could be overcome even at this concentration.

## 3. Discussion

In the quest for new sustainable approaches to control late blight, several volatile organic compounds emitted by bacteria were shown to inhibit *P. infestans* mycelial growth and development in in vitro studies, amongst which the sulfur containing S-methyl methanethiosulfonate (MMTS) [8]. MMTS proved efficient in preventing late blight symptoms *in planta* and displayed low or no phytotoxicity [8,9]. However, since older reports suggested MMTS to be fairly toxic [16], we quantified its effect on a broad range of target and non-target organisms to evaluate its suitability for agronomical application.

The results indicated that *P. infestans* is more susceptible to MMTS compared to the fungi *R. solani*, *B. cinerea*, *T. harzianum* and *T. artroviride.* The oomycete was affected at lower concentrations in mycelial growth (ED_50_ 2.4 µg/mL) than all fungal species tested. In fact, up to an MMTS concentration of 33 µg/mL, fungi were never 100% inhibited and could overcome initial inhibition after long term exposure, whereas *P. infestans* seemed permanently inhibited at 10 µg/mL MMTS. Additionally, *P. infestans* sporangia and zoospores were also impaired in germination at lower concentrations than *B. cinerea* spores. This was especially true for zoospores that showed impaired germination at concentrations 1000 times lower than *B. cinerea* spores. Higher sensitivity of zoospores was expected, since they are short-lived, cell wall-less structures. Not surprisingly, the difference in sensitivity to MMTS was less pronounced between *P. infestans* sporangia and *B. cinerea* spores, although the fungal spores were still tenfold less sensitive than the oomycete ones.

However, in comparable studies on fungicidal spectrum of cyazofamid and ethaboxam—compounds currently in use for oomycete control in the field—the difference in sensitivity between the oomycete and fungi was more pronounced than that observed for MMTS [17,18]: ED_50_ values for cyazofamid against *P. infestans* mycelial growth, for example, ranged between 0.008 and 0.02 µg/mL whilst the growth of *B. cinerea* and *R. solani* were not affected up to 100 µg/mL [17]. S-methyl methanethiosulfonate thus exerts more broad-spectrum effects than other anti-oomycete agents. This broad range of activity of MMTS is also confirmed by the growth inhibition observed for yeast and bacteria, both being inhibited at MMTS concentrations of 6–30 µg/mL.

To avoid too many experiments on mammals for early safety testing, the model nematode *C. elegans* is widely used for toxicity assessment of hazardous compounds and potential drugs [19]. Furthermore, nematodes fulfill important functions in the soil environment, for example in nutrient turnover [20]. Their communities have been shown to be affected by different soil management practices [21] as well as by application of fungicides [22]. Including toxicity assessment of MMTS on nematodes in this study was thus crucial to obtain a full picture of its potential environmental impact.

The LD_50_ value of 2 µg/mL obtained for MMTS on *C. elegans* is a further proof of its broad range of action. In comparison, dichlorvos—an organophosphorus insecticide—displayed toxicity on *C. elegans* with LD_50_ values that ranged between 0.49 µg/mL and 8.617 µg/mL [23,24]. Moreover, Li and colleagues (2013) published a correlation study between LD_50_ values in rats and *C. elegans*, to evaluate how well rodent LD_50_ values could be extrapolated from nematode LD_50_ values. Comparing their data with the LD_50_ obtained for MMTS suggests its classification in category three out of five in the Globally Harmonized System of Classification and Labeling of Chemicals (GHS) [23]. Whether such direct comparisons can be made is however debatable. Firstly, because in their study, experiments were carried out on l4-larvae instead of l1-larvae and they were exposed to test-compounds for a maximum of 24 h and not 48 h. These differences in experimental setup might lead to an overestimation of MMTS toxicity. In addition, correlation between rat LD_50_ values and *C. elegans* LD_50_ values was not always reliable and only two compounds were tested with an LD_50_ value on *C. elegans* comparable to the one obtained for MMTS [23]. Moreover, note that MMTS might function as an appetite-depressant [25] and that bacteria added as food source for the nematodes during experimentation were also affected by the compound. This renders interpretation of toxicity data on *C. elegans* additionally difficult since larval death and underdevelopment could be caused by acute toxicity or starvation. Still, a LD_50_ value of 2 µg/mL is comparably low, which is in line with the rather severe toxicity of MMTS observed in earlier studies on mice [16].

Finally, our results show that the response of plant seedlings to MMTS followed a similar pattern as fungal mycelia. Initially halted in growth at 10 µg/mL of MMTS, seedlings overcame inhibition entirely after longer growth period, suggesting only low phytotoxicity. Although symptoms remained at 33 µg/mL, even at that comparably high concentration, seedlings could tolerate the compound under laboratory conditions. Furthermore, we previously reported lower phytotoxicity of MMTS compared to dimethyl trisulfide—another sulfur containing volatile organic compound—and little to no symptoms on potato plantlets treated with MMTS-doses high enough to entirely inhibit late blight symptoms [9]. However, whether these concentrations would also be sufficient for late blight control in field-grown potatoes remains to be investigated.

Overall, our results show that while MMTS inhibited the growth of a wide range of organisms, their level of sensitivity differed and the oomycete *Phytophthora infestans* remained more sensitive than most non-target organisms. The sensitivity of the zoospores, which play a crucial role in the infection of both leaves and tubers, was particularly impressive.

The broad range of action displayed by this compound might be explained by the ability of MMTS to oxidize thiol-groups in free cysteine, cysteine in proteins and potentially also in glutathione and coenzyme A reversibly [26,27] thereby altering protein activity and pushing cells into oxidative stress [28]. The general downregulation of proteins paired with an upregulation of antioxidants observed in the *P. infestans* proteome upon MMTS treatment also supports this hypothesis [9]. Moreover, we could confirm the interaction between MMTS and thiol-groups on cysteine by demonstrating that the inhibitory effect of MMTS on bacterial growth was lost when applied together with cysteine but not with serine (Appendix A).

In addition to MMTS, interaction with thiol-groups has been proposed to underlie the antimicrobial activity of other compounds. A prime example is allicin, which has been extensively studied for its antimicrobial potential [28]. Additionally, dimethyl disulfide (DMDS), a compound derived from the same pathway as MMTS [10,11], is thought to exert toxic effects in the same manner [25,29,30] and has been linked to poisoning and hemolytic anemia in ruminant species after feeding on *Brassica* plants [25,29,30].

In addition to its high reactivity with thiol-groups, MMTS interacts with different sites in bio-membranes indicating that it might easily diffuse over and destabilize membranes [31,32]. This could have added to the particular susceptibility of cell-wall-less zoospores that appeared wrinkly after MMTS treatment. We thus propose that differential susceptibility of organisms to MMTS mainly depends on permeability, i.e., cell wall composition and membrane composition, number of functional cysteine groups within proteins as well as sensitivity to oxidative stress. With that in mind, specifically targeting *P. infestans* with MMTS might be a good strategy, since *P. infestans* encodes for numerous secreted cysteine-rich effectors (virulent factors) important for successful infection of the host [33,34,35].

In addition to the overrepresentation of cysteine in secreted proteins, there is also an interesting positive link between the complexity of an organism and the amount of cysteine in proteins [33], which would imply that higher organisms could be more susceptible to MMTS. Higher amount of cysteine in proteins however, is thought to be accompanied by a higher amount of reducing agents, keeping cysteines in cytosolic proteins in the thiol-form [33,36]. Plant thioredoxin systems have been described to be particularly elaborate, which could at least partially explain why *A. thaliana* seedlings tolerated MMTS treatment comparably well [37].

In contrast to toxic effects observed at high doses, MMTS might have health promoting characteristics in low doses. As suggested for allicin, thiol-modifying agents can act as antioxidants in low doses by boosting cellular response to oxidants [28]. Interestingly, the biosynthetic precursor of MMTS in plants, S-methyl-cysteine has been proposed to exhibit antioxidant function. Furthermore, Nakamura and colleagues observed reduced UV mutagenesis after MMTS treatment and linked thiol-modifying capacity of MMTS with activation of the excision repair system in *E. coli* [11].

In conclusion, our results as well as findings of other studies conducted on MMTS or related compounds, suggest that MMTS is likely too toxic to consider for application as pure compound in the field. However, it might still be useful for pathogen control, especially in view of its very general mode of action, which, similarly to copper, might prevent resistance emergence. In analogy to the more toxic hydrogen cyanide—which is a key component of the biocontrol efficiency of *Pseudomonas* strains [38,39]—MMTS occurring in a blend of naturally produced volatiles emitted by the plant microbiome could be a powerful component to suppress plant diseases, especially diseases caused by *P. infestans* or other *Phytophthora* species.

## 4. Materials and Methods

### 4.1. Biological Material, Culture Conditions and Reagents

If not mentioned otherwise *P. infestans* strain 88069 provided by Prof. Francine Govers (Wageningen University, Netherlands) was used for all experiments. In some cases, *P. infestans* strain Rec01 [7] was used additionally to 88069. *P. infestans* strains were kept in the dark at 18 °C as mycelial cultures on 10% V8-Agar medium [40] supplemented with 0.1% calcium carbonate. They were regularly transferred to potato tubers to maintain their virulence.

*Trichoderma artroviridea* strain M134 and *T. harzianum* strain M136 were kindly provided by Dr. Saskia Bindschedler (University of Neuchâtel, Switzerland), *B. cinerea* strain BMM by Prof. Brigitte Mauch-Mani (University of Neuchâtel, Switzerland) and *R. solani* strain 5900 was provided by Syngenta. All fungal strains were routinely grown on 3.9% Potato Dextrose Agar (PDA, Sigma-Aldrich) and incubated—with the exception of *R. solani*—at 20 °C with a 12h light/ 12h dark cycle. *R. solani* was incubated in the dark at 18 °C.

Yeast *S. cerevisiae* strain BY4741 was provided by Prof. Roger Schneiter (University of Fribourg, Fribourg, Switzerland) [41]. It was grown on synthetic minimal glucose medium (SD), which was prepared from 0.68% Yeast Nitrogen Base without amino acids (Sigma). Zero-point two percent glucose, amino acids and—for culture on plates—Agar-Agar Kobe I (Roth) were added to the medium. To prepare liquid yeast cultures in exponential growth phase, 2–3 colonies were suspended in SD medium and grown to stationary phase for 48 h at 30 °C and 190 rpm. Part of this stock culture was then diluted 1000× in SD and grown overnight in the same conditions as before. *Dickeya solani* strain 07/044 was provided by Dr. Santiago Schaerer (Agroscope, Nyon, Switzerland). Bacterial cultures were grown on Lysogeny Broth (LB) medium at room temperature. LB medium was prepared by mixing 12.5 g/L Millers Luria, 10 g/L Lennox and for solid culture 15 g/L agar-agar in distilled water.

*A. thaliana* seeds (ecotype Col-0) were sterilized by immersion in 70% ethanol for 30 min and in 100% ethanol (absolute., ≥ 99.8%, Fisher Chemical) for 1 min. They were dried under a sterile hood and stored at 4 °C. *A. thaliana* seedlings were grown on ½ Murashige and Skoog (½ MS) 0.8% agar medium (2.165 g/L Murashige and Skoog Basal Salt Mixture, 1 mL/L vitamin stock solution) (Sigma-Aldrich) supplemented with 1% glucose.

All media were prepared with distilled water and sterilized by autoclaving 20 min at 120 °C.

S-methyl methane thiosulfonate (MMTS) was ordered from Sigma (n°64306) and stored at 4 °C. If not mentioned otherwise serial dilutions of MMTS were prepared in dimethyl sulfoxide (DMSO, Acros Organics, part of Thermofischer Scientific, Waltham, MA, USA). Pure DMSO was used as solvent control.

### 4.2. Effect of MMTS on Mycelial Growth and A. thaliana Seedlings

The effect of MMTS on mycelial growth of the fungi *B. cinerea*, *R. solani*, *T. harzianum, T. atroviridae* as well as the oomycete *P. infestans* was assessed by placing a 5 mm agar plug—removed from the edge of an actively growing mycelium—upside down in the middle of a standard Petri dish (Greiner bio-one). Petri dishes contained appropriate growth medium (PDA and V8 for fungi and *P. infestans* respectively) supplemented with MMTS at concentrations ranging from 1–33 µg/mL (0.006 vol% DMSO). Plates were sealed with parafilm, placed upside down in plastic bags and incubated at 21 °C in the dark. Growth progression was monitored at two different timepoints by taking photographs. Timepoints depended on the growth-speed of the fungi or oomycete. The first timepoint was set shortly before the mycelium of DMSO control plates reached the edge of the Petri dish and corresponded to 3–4 and 7 days for fungi and *P. infestans,* respectively. A second timepoint was examined after control plates were fully overgrown, which corresponded to 10 and 14 days for fungi and *P. infestans*, respectively. Mycelial growth areas were later measured in cubic centimeters using the lasso selection tool and spatial calibration function in ImageJ software. These experiments were repeated twice with four replicates per repetition.

To assess the effect of MMTS on *A. thaliana* seedlings, ecotype Columbia (Col-0) seeds were sown and left for germination on square Petri dishes (Greiner bio-one) containing half MS medium with Glucose. Four days post germination, seedlings were transferred to new square plates with the same medium but with added MMTS at final concentrations ranging from 1–33 µg/mL (0.006 vol%). DMSO (0.006 vol%) as well as a water controls were included in the experiment. Six to seven seedlings were placed per plate, which were sealed with parafilm and incubated vertically at 21 °C with a 12 h photoperiod. Primary root length was measured from pictures taken 5 days after the transfer using the simple neurite tracer and spatial calibration functions in ImageJ. Further growth progression was followed by taking pictures. This experiment was repeated twice.

### 4.3. Effect of MMTS on P. infestans Zoospores, Sporangia and B. cinerea Spores

To collect *P. infestans* zoospores and sporangia 2–3 weeks old *P. infestans* mycelial cultures grown on V8 agar medium were used. For zoospores, the mycelium was covered with ice-cold sterile water and incubated at 4 °C in the dark for 2 h. To trigger zoospore release, the plates containing the mycelium were subsequently placed at room temperature for 20 min in the dark. Freely swimming zoospores were then collected by pipetting the upper layers of water covering the mycelium. Zoospore density was adjusted to approximately 100,000–500,000 zoospores/mL. To harvest *P. infestans* sporangia, 1 mL of sterile water was added to the mycelium, which was then scraped off the growth medium and transferred to a Falcon tube. To detach the sporangia from the mycelium the tube was thoroughly agitated but not vortexed. The resulting turbid liquid containing sporangia was gathered by filtering the mycelium-solution through a mesh. Sporangia concentration was adjusted to approximately 5000 sporangia/mL.

To harvest spores from *B. cinerea* grown on PDA, the mycelium was collected in sterile water and filtered through glass wool. The liquid was subsequently centrifuged at 800 rpm and 4 °C for 10 min and the pellet resuspended in 500 µL of sterile water. Finally, the spores were diluted in PDB ¼ to a density of 500,000 spores/mL.

To test the effect of MMTS on *P. infestans* zoospores, sporangia and *B. cinerea* spores, MMTS was added to final concentrations of 500 pg/mL, 5 ng/mL, 50 ng/mL, 500 ng/mL, 5 µg/mL, 50 µg/mL, 100 µg/mL and 500 µg/mL (1 vol%) in 24 well plates (Corning Costar n°3526) containing 200 µL of either spore suspension. Only selected concentrations were tested for each type of spore. Plates were sealed with parafilm and water-agar (1%) was poured into the lid to prevent transfer of volatile MMTS from one well to another. *P. infestans* zoospores were then immediately imaged in order to see the effect of the compound on motility. Motility was judged as normal when the large majority of zoospores were observed to swim freely 5–15 min post treatment and judged impaired when no or only single zoospores were observed to move within this time-window. After 30–40 min propidium iodide (PI) was added to half the wells with zoospores and sporangia to assess the number of dead cells per treatment. The other half was left to germinate for another 16–18 h. *B. cinerea* spores were incubated with MMTS for 8 h at 20 °C prior to first imaging and imaged again after 24 h to see progression of germination. All imaging was done using the Cytation 5 (Biotek) and image analysis was performed in ImageJ.

### 4.4. Effect of MMTS on D. solani in Comparison to Kanamycin

To assess the effect of MMTS on bacterial growth in comparison to kanamycin, a *D. solani* overnight liquid culture was diluted in test tubes to an OD_595_ = 0.05. MMTS or kanamycin were added to the tubes to obtain final concentrations of 3–300 µg/mL for MMTS and 3–30 µg/mL for kanamycin. DMSO—solvent for MMTS—and a water—solvent for kanamycin—controls were included in the experiment. Bacteria were incubated at 28 °C and 180 rpm. OD_595_ was measured after 4 h, 7 h, 22 h and 28 h.

### 4.5. Effect of MMTS on Yeast Growth

*S. cerevisiae* liquid SD culture in exponential growth phase (OD_600_ = 0.3) was diluted to OD_600_ = 0.05 with SD medium. One mL of this diluted culture was added per well in a 24 well plate (Corning Costar n°3526). MMTS was then added to final concentrations of 1.95–250 µg/mL (0.2 vol%). A non-treated as well as a DMSO control (0.2 vol%) were included in the experiment. Water agar (1%) was poured into the space surrounding the wells to create barriers for volatile MMTS exchange between the wells. Plates were sealed with parafilm and incubated in the Cytation 5 (Biotek) plate-reader at 30 °C while shaking at 237 rpm for 24 h. OD_600_ was measured every 30 min to obtain growth curves.

### 4.6. Effect of MMTS on C. elegans Larvae

The toxicity of the MMTS compound was evaluated on synchronized L1 larvae of the model nematode *C. elegans*. The worms and eggs were collected from Nematode Growth Medium (NGM) plates with sterile water in 1.5 mL Eppendorf tubes. They were then centrifuged at 1200× *g* for 2 min to precipitate the worms and embryos. The pellet was washed twice with sterile water to get rid of bacterial debris. To isolate embryos from the hatched worms, one ml of sterile water contained final concentration of 0.6% sodium hypochlorite and 100 mM KOH was added to the pellet. The solution was incubated 10–12 min at room temperature with shaking (1200 rpm) in a shaker incubator block (Thermo Mixer 1.5 mL, Eppendorf, Germany). After dissolving worms in the bleach solution, the tube was centrifuged at 1200× *g* for two min to precipitate embryos. Next, the pellet was washed twice with distilled water and Phosphate Buffer Saline (PBS) pH: 7.4. The cleaned eggs were incubated in the NGM plates for 12 h at 20 °C in the dark. Subsequently, the synchronized L1 larvae were collected with sterile water from the plate and washed twice with PBS buffer.

Approximately 50 L1 larvae were exposed to 1 µg/mL–1 mg/mL MMTS in 96 well plates (Corning Costar n° 3596). The wells contained living *E. coli* OP50 in PBS pH: 7.4 buffer at final OD_600_: 1, as a food source. The plate was sealed with parafilm and incubated at 20 °C for 48 h in the dark. A non-treated well and a DMSO (2%) were considered as controls. Finally, living worms were counted per well to obtain LD_50_ values. The experiment was performed twice and comprised six technical replicates for each treatment.

### 4.7. Data Analysis

All data analysis was carried out in RStudio [42] and Microsoft Excel software. LD_50_ and ED_50_ values were determined by fitting a log-logistic model to the data with the “dcr” (Dose Response Curve) R-package [43], growth curve parameters were obtained using the “growthcurver version 0.3.0” R-package (https://rdrr.io/cran/growthcurver/). Whenever the data allowed it, a four parametric log-logistic model was used. If not possible, a three parametric model was used instead. Graphs were realized with the help of ggplot2 R-package [44]. Significant differences between treatments were determined by one-way ANOVA followed by Tukeys Honest Significant Difference (HSD; significance levels between all treatments) or Least Significant Difference (LSD; grouping of treatments) for post-hoc analysis. Data from independent experiments were pooled for statistical analysis only if multi-way ANOVA showed no or negligible effect for the factor “experiment”.

## Figures and Tables

**Figure 1 pathogens-09-00496-f001:**
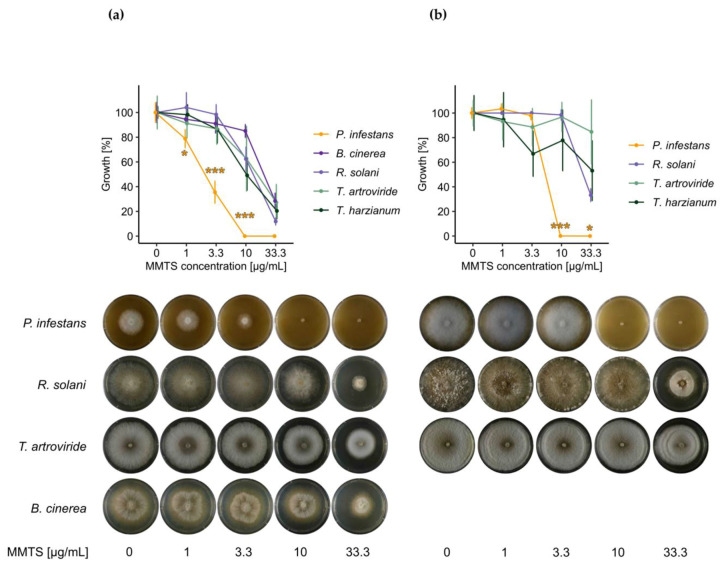
Inhibition of mycelial growth of *P. infestans* grown on V8 medium and *R. solani*, *T. artroviride*, *T. harzianum*, *B. cinerea* grown on Potato Dextrose Agar (PDA) medium at methanethiosulfonate (MMTS) concentrations of 1–33 µg/mL. Representative pictures and corresponding growth area quantifications in percentage of the solvent control are shown at an early timepoint (**a**) and a later timepoint (**b**). Timepoint 1 corresponded to 3–7 days after inoculation, Timepoint 2 to 10–14 days after inoculation (depending on mycelial growth speed). Average growth compared to the solvent control (100%) depicted in the upper graphs corresponds to the mean of eight replicates pooled from two independent experiments with four replicates each. Error bars represent standard deviation (*n* = 8). Significant differences in growth percentages at a given MMTS concentration between *P. infestans* and all fungal species are indicated by yellow asterisk (one-way ANOVA and Tukey’s HSD for post-hoc analysis; significance levels: *p*-values: ≤0.001 ***, ≤0.05 *). Significant differences between fungal species are depicted in Appendix A.

**Figure 2 pathogens-09-00496-f002:**
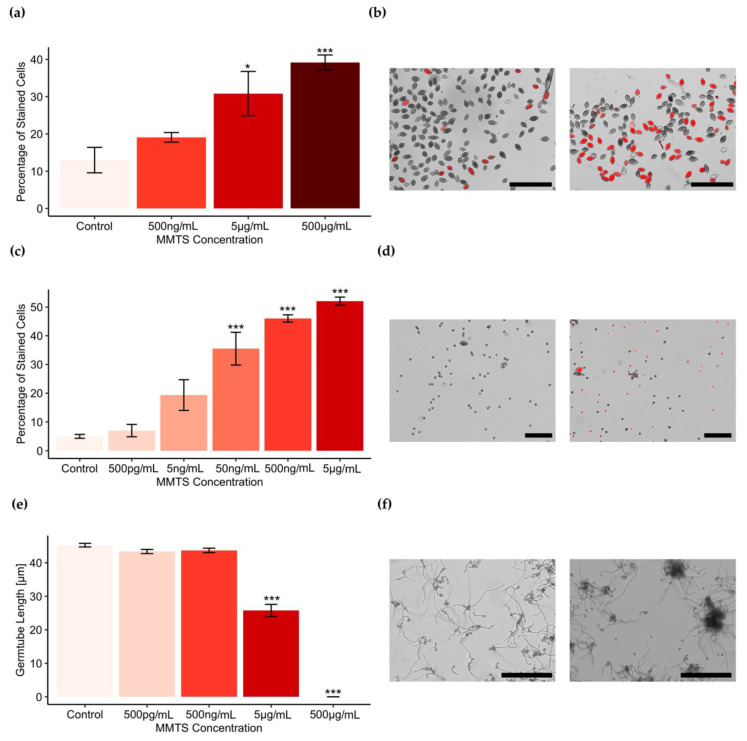
MMTS treatment of *P. infestans* sporangia (**a**,**b**) and zoospores (**c**,**d**), as well as *B. cinerea* spores (**e**,**f**); (**a**–**d**), *P. infestans* sporangia and zoospores were incubated with MMTS at various concentrations for 30 min and thereafter stained with propidium iodide; (**a**,**c**) bars represent average percentage of stained cells per treatment of six replicates with standard error; (**b**) representative pictures of sporangia stained with propidium iodide after treatment with solvent control (left panel) or 500 µg/mL of MMTS (right panel). Black bars correspond to 200 µm; (**d**) representative pictures of zoospores stained with propidium iodide after treatment with solvent control (left panel) or 5 µg/mL of MMTS (right panel). Black bars correspond to 200 µm; (**e**) *B. cinerea* spores. Bars represent average germ tube length after 8 h incubation of 5 replicates of *B. cinerea* spores treated with 500 pg/mL–500 µg/mL MMTS; (**f**) representative pictures of germinated *B. cinerea* spores incubated for 24 h in the solvent control (left panel) and exposed to 5 µg/mL MMTS (right panel). Black bars represent 200 µm. Asterisks in graphs represent statistically significant differences in comparison to the control (ANOVA and Tukey’s HSD for post hoc analysis; significance levels: *p*-values: ≤0.001 ***, ≤0.05 *).

**Figure 3 pathogens-09-00496-f003:**
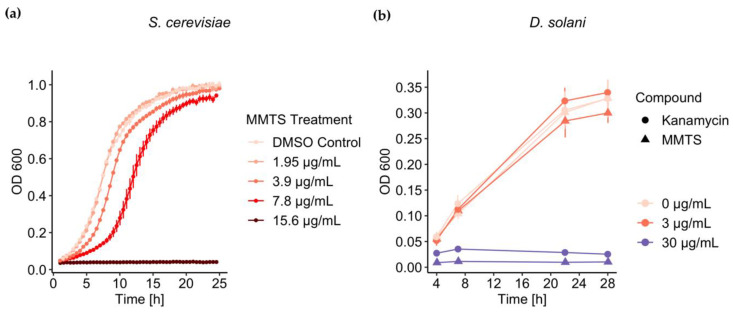
(**a**) Growth curves of *S. cerevisiae* exposed to 1.95–15.6 µg/mL MMTS in liquid culture. Points represent average OD_600_ of three replicates at a given time point and concentration. Error bars correspond to standard error. Significant differences between growth curves are represented in Appendix A; (**b**) growth curve of *D. solani* exposed to 0, 3 or 30 µg/mL of MMTS in comparison to kanamycin at the same concentrations. Symbols indicate average OD_595_ for a given time point, concentration and compound of 3 replicates. Error bars represent standard error. No significant differences between growth curves of bacteria treated with kanamycin or MMTS could be observed (Appendix A).

**Figure 4 pathogens-09-00496-f004:**
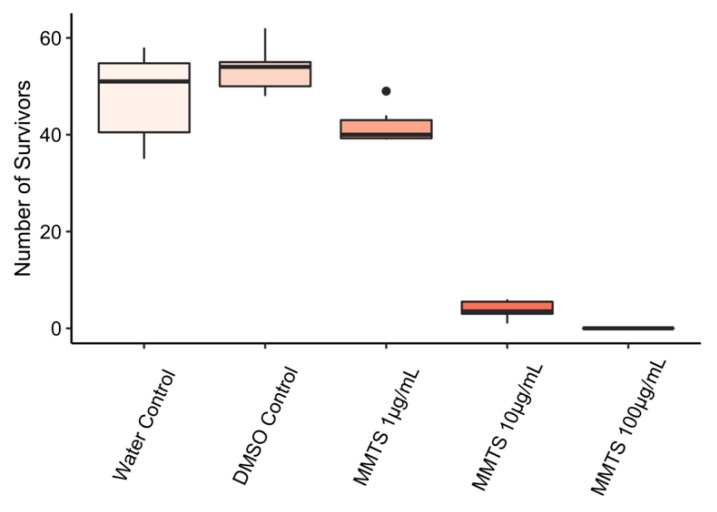
Survival of *C. elegans* L1 larvae exposed to 1–100 µg/mL MMTS for 48 h. Thick lines in boxplots indicate the average of six replicates. Different letters show significant differences in survival between the treatments according to ANOVA and Tukey’s HSD for post hoc analysis (*p* < 0.05). In samples treated with a concentration of 10 µg/mL of MMTS and above no adult worms were visible. The experiment was repeated twice with similar results.

**Figure 5 pathogens-09-00496-f005:**
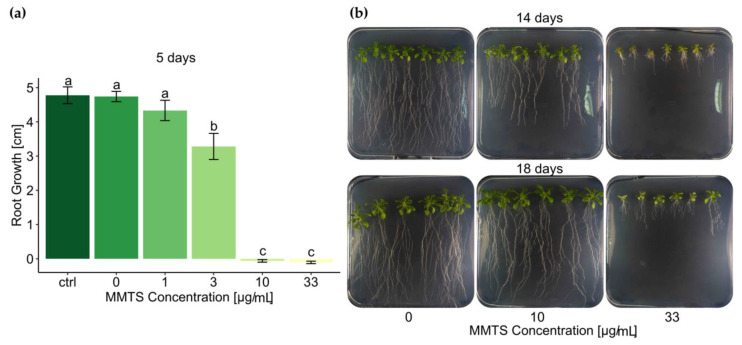
Growth inhibition of *A. thaliana* seedlings by MMTS: (**a**) bars show average root growth of seedlings after five days of incubation on growth medium with 0–33 µg/mL MMTS with standard error. The experiment was carried out in 3 repetitions with 6–7 seedlings each. Letters indicate significant differences in root growth between treatments according to ANOVA and Tukey’s HSD for post hoc analysis (*p* < 0.05); (**b**) representative pictures of seedlings after growing for 14 days (upper panel) and 18 days (lower panel) on medium containing 0, 10 or 33 µg/mL MMTS.

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
