# Peer review of "S-methyl Methanethiosulfonate: Promising Late Blight Inhibitor or Broad Range Toxin?"

_pathogens, 2020, doi:10.3390/pathogens9060496_

Round 1

Reviewer 1 Report

Review of Manuscript ID: pathogens-825682

The manuscript aims to test the toxicity of S-methyl Methanethiosulfonate (MMTS), a volatile organic compound produced by bacteria that are associated with potato both to Phytophthora infestans (causal agent of late blight of potato), to other pathogen and non-pathogenic fungi and oomycetes, as a well as a model nematode and model plant. The work builds on previous work by some of the current authors in which MMTS was found to have potential against P. infestans. The rationale for the work is well explained and put into a broader context in the introduction.

The choice of fungi, oomycetes, yeast, bacteria and Arabidopsis and C. elegans to test are logical and the methods of testing are generally logical and appropriate.

The results are generally well presented and explained. It would be helpful to have an assessment of statistical significant in Figure 1, instead of only in Figure S1. It took several readings of the legend of Figure 2 to figure out that panels (e) and (f) correspond to Botrytis cinerea. This should be clarified.

The first paragraph of the discussion repeats the introduction to some extent and should be edited or removed.

The discussion of interaction between MMTS and thiol-groups in cysteine as a potential explanation for the mode of action of MMTS and figure S5 was particularly interesting, insightful and thorough.

The conclusions drawn are logical and reasonable, given the results.

Minor editorial comments:

Abstract:

Remove duplication of the word “nevertheless” in line 21

Introduction:

Remove the word “and” in line 30 and replace with a comma

Replace word “attributed” in line 46 with e.g. “devoted”

Abbreviation MMTS not used in lines 55-56

Discussion:

Avoid one-sentence paragraphs (lines 247-248).

Materials and Methods:

How were photographs of the growing cultures translated into distances for growth measurements? (lines 370 – 371). Later in the paragraph, ImageJ is mentioned. This could be clarified.

Remove the word “A” before DMSO on line 380

How were the photographs of Arabidopsis translated into root length? (lines 383-384)

General comment: Generally well written. Tendency to overly long sentences that would benefit from be divided in two.

Reviewer 2 Report

I found this to be a very well-written and well-presented piece of research detailing broad-spectrum effects of the compound MMTS. The topic is well introduced, methods and results are clearly explained, and the results are put nicely into the wider context of disease prevention along with a concise discussion of the compounds possible mode-of-action.

The examination of P. infestans life stages and their differing response to MMTS is interesting and relevant to real-world infection. While the compound is shown to affect a broad range of organisms, the observation that infestans has increased sensitivity, particularly at the zoospore stage, suggests that MMTS does warrant some further investigation. The authors acknowledge this in the discussion, with their honest assessment that pure compound application is likely to have toxicity issues, but that the compound may have merit in a naturally-produced mixture.

The statistics in Figure 1 may need to be revisited - two independent experiments with 4 replicates each would constitute n=2 rather than n=8, with the plate replicates in each being 'technical replicates'.

Author Response

The statistics used for figure 1 have now been clarified in the legend: we performed an ANOVA to check whether the two independent experiments differed singificativeyl, and since this was not the case, we pooled all replicates prior average, yielding n=8. This information has been added to the legend of Figure 1.